# Dynamic Early Terminating of Multiply Accumulate Operations for Saving Computation Cost in Convolutional Neural Networks

## Abstract

Deep learning has been attracting enormous attention from academia as well as industry due to its great success in many artificial intelligence applications. As more applications are developed, the need for implementing a complex neural network model on an energy-limited edge device becomes more critical. To this end, this paper proposes a new optimization method to reduce the computation efforts of convolutional neural networks. The method takes advantage of the fact that some convolutional operations are actually wasteful since their outputs are pruned by the following activation or pooling layers. Basically, a convolutional filter conducts a series of multiply-accumulate (MAC) operations. We propose to set a checkpoint in the MAC process to determine whether a filter could terminate early based on the intermediate result. Furthermore, a fine-tuning process is conducted to recover the accuracy drop due to the applied checkpoints. The experimental results show that the proposed method can save approximately $50\%$ MAC operations with less than $1\%$ accuracy drop for CIFAR-10 example model and Network in Network on the CIFAR-10 and CIFAR-100 datasets. Additionally, compared with the state-of-the-art method, the proposed method is more effective on the CIFAR-10 dataset and is competitive on the CIFAR-100 dataset.

## 1 Introduction

In recent years, deep learning with deep neural network (DNN) has been a breakthrough solution in machine learning. Its success has been demonstrated in many artificial intelligence (AI) applications, such as image classification (Krizhevsky et al. (2012)), face recognition (Learned-Miller et al. (2016)), object detection (Redmon et al. (2016)), big data analysis (Gheisari et al. (2017)), medical science (Litjens et al. (2017)), etc. Due to the high computational complexity, a DNN is usually implemented on a powerful hardware device, such as a graphics processing unit workstation. However, considering the diversity of AI applications, it is desirable to implement a neural network model on an edge device which usually has limited hardware resources for saving power and cost.

To this end, many model compression and acceleration methods were proposed to compress neural network models. Their common objective is to reduce the computational complexity and/or memory usage without significantly affecting the performance and accuracy. The existing methods have been classified into four categories (Cheng et al.): (1) parameter pruning (Han et al. (2015a); Han et al. (2015b)), (2) low-rank factorization (Jaderberg et al. (2014); Lebedev et al. (2014); Tai et al. (2015)), (3) convolutional filter transformation (Szegedy et al. (2017)), and (4) knowledge distillation (Hinton et al. (2015)). The parameter pruning-based methods eliminate noncritical parameters for simplifying models. The low-rank factorization-based methods exploit low-rank matrix approximation to simplify convolutional filters. The convolutional filter transformation-based methods transform filters based on equivariance properties to compress convolutional neural network (CNN) models. The knowledge distillation-based methods train a compact neural network to mimic the function of a large network with the distilled knowledge from the large network.

Most compression methods fixedly modify the structure of a model. However, the criticality of a parameter or a filter heavily depends on the input data. A critical filter may become non-crucial under different input cases. These static methods may not be comprehensive enough when con-

sidering diverse input data. Thus, in this paper, we propose a dynamic optimization method for CNNs. The proposed method does not modify the structure of a model, but introduces a specific multiply-accumulate (MAC) unit to reduce the MAC operations dynamically in the inference phase.

In a CNN, a convolutional layer is usually followed by an activation layer and a pooling layer. The activation layer introduces non-linearity into the system. Rectified linear unit (ReLU) is the most popular activation function, which applies $f(x) = max(0, x)$ to rectify the feature maps from the convolutional layer. Since negative values are changed to $0$, the computation efforts of the filters that output negative values in the convolutional layer can be seen as wasteful. Furthermore, a pooling layer reduces the scale of the feature maps by extracting only significant data. The most common pooling method, max pooling, preserves the data with the largest value. Similarly, if the output value of a convolutional filter is not large enough to pass through the max pooling layer, the computation efforts of the filter are wasteful as well.

Based on the observations, if we are able to detect in advance that a convolutional filter will output a negative value, we can stop it early for saving computation efforts. A convolutional filter basically conducts a series of multiply-accumulate (MAC) operations. Since the output is an accumulated result, it could be feasible that using an intermediate result in the MAC process to predict the sign of the output. Thus, we propose to determine whether a convolutional filter should continue or terminate according to an intermediate result.

In the proposed optimization method, a convolutional filter is given a checkpoint. When the checkpoint iteration is reached, we check the intermediate result of the MAC process. If the result is less than $0$, we stop the MAC process to save the remaining operations and output the intermediate result directly; otherwise, the MAC process finishes all the MAC operations. Since saving the MAC operations may affect the accuracy, we further fine-tune the optimized CNN to improve its accuracy with the applied checkpoints.

Unlike the previous static optimization methods, which explicitly compress a CNN before the inference phase, the proposed method is dynamic. It determines the saved computation efforts during the inference phase, and the saved efforts vary for different input data. Furthermore, unlike the previous methods (Viola & Jones (2001); Teerapittayanon et al. (2016)) that terminate the whole network and make a prediction directly at an early layer, the proposed method focuses on early stopping the MAC processes in the convolutional filters. Basically, each input data still needs to go through the whole network for making the prediction.

In the experiments, we apply the proposed method to CIFAR-10 example model[1] (Jia et al. (2014)) and Network in Network (Lin et al. (2013)), with the CIFAR-10 and the CIFAR-100 datasets (Krizhevsky (2009)). For CIFAR-10 example model, the results show that our method saves $50.22\%$ MAC operations with only $0.8\%$ accuracy drop under the CIFAR-10 dataset, and saves $43.64\%$ MAC operations with only $0.09\%$ accuracy drop under the CIFAR-100 dataset. For Network in Network, our method saves $47.43\%$ MAC operations with only $0.41\%$ accuracy drop under the CIFAR-10 dataset, and saves $47.73\%$ MAC operations with only $0.58\%$ accuracy drop under the CIFAR-100 dataset. Furthermore, our method is competitive with the existing methods (He et al. (2017); Li et al. (2016)). For the CIFAR-10 dataset, our method even saves more MAC operations with less accuracy drop.

In summary, we propose a dynamic optimization method for CNNs, which reduces MAC operations in the inference phase. It effectively saves the computation cost of a CNN with only a small effect on the accuracy. Additionally, it allows a user to trade off the saved computation cost against the accuracy drop.

The remainder of this paper is organized as follows: Section 2 reviews some background and related works. Section 3 shows a motivational example and illustrates the key idea behind the proposed method. Section 4 presents the proposed method. Section 5 shows the experimental results. Finally, the conclusion is presented in Section 6.

---

[1]The CNN model is from `http://caffe.berkeleyvision.org/gathered/examples/cifar10.html`

## 2 PRELIMINARIES

### 2.1 CNN AND CONVOLUTIONAL FILTER

CNNs have been extensively used in many AI fields, such as computer vision and natural language processing. A CNN is a multi-stage model. In general, a stage is composed of a convolutional layer, an activation layer and a pooling layer. The convolutional layer applies convolution operations to the input feature maps. The following activation and pooling layers rectify the feature maps and reduce their scale, and pass the outputs to the next stage. Additionally, one or multiple fully connected layers are applied to the last few layers of a CNN.

Convolutional layers are the core of a CNN. A convolutional layer is made up of a set of learnable filters and a filter is composed of weights and a bias. The functionality of a filter at the $l^{th}$ layer can be mathematically formulated as EQ 1, where, $K$, $W$, and $H$ are the depth, width, and height of the filter, respectively. $z_{p,q,k}^{l-1}$ is an input from the $l-1^{th}$ layer and $h_{p,q,k}$ is the corresponding weight in the filter. Additionally, $b$ is the bias. There are a total of $H \times W \times K + 1$ MAC operations for the filter to generate an output value. For convenience, we call the process of conducting the $H \times W \times K + 1$ MAC operations a *MAC process*. Depending on the size of the input feature maps, a filter usually needs to conduct many MAC processes for generating the outputs.

$$u = \sum_{k=0}^{K-1} \sum_{q=0}^{W-1} \sum_{p=0}^{H-1} z_{p,q,k}^{l-1} h_{p,q,k} + b \tag{1}$$

Basically, the convolutional layers dominate the computation efforts of a CNN. Thus, we propose to reduce the MAC operations for saving computation efforts.

### 2.2 RELATED WORKS

In recent years, many efforts have been devoted to simplifying the computational complexity of CNNs. Most methods aim to simplify a CNN model by removing redundancies or noncritical parameters without affecting the overall accuracy.

The low-rank factorization-based methods approximate convolution operations by decomposing a weight matrix into a product of low-rank matrices (Jaderberg et al. (2014); Lebedev et al. (2014); Tai et al. (2015); Sainath et al. (2013); Kim et al. (2015); Chung & Shin (2016)). As a result, the convolutional layers are simplified, but the filter count does not change.

Pruning parameters was initially to prevent a trained model from overfitting to the training dataset (Hassibi & Stork (1993); LeCun et al. (1990)). However, researchers then observed that there may exist redundant or noncritical parameters in a model. Thus, some works propose to eliminate connections between layers by pruning small-magnitude weights (Han et al. (2015a); Han et al. (2015b); Han et al. (2016)). However, the simplification does not necessarily reduce computation time and the irregular sparsity in the convolutional layers requires specific implementation.

Instead of pruning weights, recent methods prune filters directly without introducing irregular sparsity (Li et al. (2016); He et al. (2017)). Li et al. propose to prune filters that are identified as noncritical to the output accuracy (Li et al. (2016)). The method significantly reduces the computation efforts by removing whole filters and the corresponding feature maps. Furthermore, instead of analyzing filter weights, the state-of-the-art method proposed by He et al. exploits redundancies in feature maps to prune filters and feature maps, and reconstruct the following feature maps to the next layer (He et al. (2017)).

Unlike the previous methods, the proposed method does not explicitly modify the structure of a model. Since the criticality of a weight/filter may differ for different input feature maps, the proposed method dynamically identifies the filters to be optimized based on the input feature maps. Additionally, a weight fine-tuning process is used to avoid significant accuracy drop.

Table 1: Motivational example. Analysis of early terminating the filters at the first convolutional layer of CIFAR-10 example model.

| $i^{th}$ iteration | 3 | 7 | 15 | 22 | 30 | 37 | 45 | 52 | 60 | 67 |
|---|---|---|---|---|---|---|---|---|---|---|
| $MAC_{TN}(\%)$ | 62.74 | 67.47 | 71.95 | 75.95 | 78.18 | 82.63 | 86.63 | 91.78 | 95.44 | 98.46 |
| $MAC_{SV}(\%)$ | 45.45 | 43.27 | 38.53 | 33.44 | 28.93 | 23.66 | 19.07 | 13.80 | 9.19 | 3.93 |

## 3 MOTIVATIONAL EXAMPLE

In a convolutional layer, a filter receives inputs from the previous layer and conducts a series of MAC operations to generate outputs to the next layer. If an output value is not large enough, it could be blocked by the following layers, and thus is not responsible for the CNN output. The key idea of the proposed method is to use an intermediate result of a MAC process to determine whether the process should stop and output the intermediate result directly, or it should complete the computation. If the intermediate result is small enough, the output value has a higher probability to be blocked and terminating the process can save MAC operations with little effect on the CNN output.

We conducted a simple experiment on the first convolutional layer of CIFAR-10 example model (Jia et al. (2014)) with 1000 input images from the CIFAR-10 dataset (Krizhevsky (2009)) to explore the feasibility of the idea. The first convolutional layer has 32 filters and each has a size of $5 \times 5 \times 3$. A filter's MAC process consists of 76 MAC operations, i.e., 76 iterations. The extra 1 is for the following bias. For each filter, we sort the weights in the decreasing order of magnitudes, so that the MAC operation with a larger-magnitude weight is computed first. Table 1 shows the probability that MAC processes with a negative intermediate result at the $i^{th}$ iteration truly output a negative value ($MAC_{TN}$), and the percentage of saved MAC operations if the processes with a negative intermediate result are terminated accordingly ($MAC_{SV}$).

As expected, $MAC_{TN}$ increases and $MAC_{SV}$ decreases as $i$ increases. That is, if we decide whether a MAC process should stop or not at an earlier iteration, we can save more MAC operations, but we have a lower probability to correctly terminate the process. However, from Table 1, it is observed that even we make a decision at the $3^{rd}$ iteration, the probability that the decision is correct is more than $60\%$. The probability is acceptable, especially when we consider the possibility that the effect of a wrong decision might be blocked by the following layers. Therefore, the strategy of early terminating a MAC process based on an intermediate result should be promising.

Next, our objective is to develop a method to determine at which iteration the intermediate result should be considered for maximizing the saved MAC operations with minimal accuracy drop.

## 4 PROPOSED METHOD

The proposed method saves MAC operations by taking the advantage of the fact that the computations in some filters could be wasteful, since the results might be set to 0 or blocked by the following activation or max pooling layers. In the following subsections, we first present the procedure of the proposed two-step method. Then, we analyze the method and discuss the ideas behind it.

In the first step, we determine the checkpoint of each filter, i.e., the iteration at which we check the intermediate result. Then, in the second step, we fine-tune the model with the applied checkpoints. After optimization, each filter has a checkpoint. During the inference phase, when a MAC process reaches the checkpoint iteration, we check the intermediate result. If it is less than 0, we terminate the process and the filter directly outputs the intermediate result; otherwise, it completes the MAC process.

### 4.1 STEP 1: SORT WEIGHTS AND SET UP CHECKPOINTS

Given a pre-trained CNN model, we first sort the weights of each filter in the decreasing order of magnitudes. Then, we determine the checkpoints based on a user-defined parameter $e_t$, which specifies the allowed maximum accuracy drop for the training dataset before Step 2. To set a checkpoint, we need to test all the training data for estimating the accuracy drop it causes. It would be very time-expensive to set the checkpoint of each filter one by one, especially for a large network with

numerous filters. Thus, we use a compromise method that determines the checkpoints layer by layer. That is, all the filters in the same layer share a common checkpoint.

We iteratively select a convolutional layer and determine its checkpoint starting from the center to the outer layers. Additionally, for two outer layers that have the same distance from the centre, we first consider the layer nearer the input. For each selected layer in which a MAC process of a filter has $n$ MAC iterations, the checkpoint could be the $\lfloor n * 5\% \rfloor^{th}$ iteration or the $\lfloor n * 32\% \rfloor^{th}$ iteration, or there is no checkpoint. We first try the checkpoint of the $\lfloor n * 5\% \rfloor^{th}$ iteration. If the resultant accuracy drop for the training data exceeds $e_t$, we then try the $\lfloor n * 32\% \rfloor^{th}$ iteration. However, if the resultant accuracy drop still exceeds $e_t$, we do not set up a checkpoint to the layer and then consider the next layer. The pseudo code of the process is shown in Algorithm 1.

---

**Algorithm 1** Accuracy-aware checkpoint setup

---

**Input:** Pre-trained CNN model $C$ with $n$ convolutional layers $L$, training data $D$, and tolerable accuracy drop $e_t$
**Output:** Optimized CNN model with checkpoints $P = \{p_1, p_2, ..., p_n\}$
 1: $acc \leftarrow$ measure accuracy of $C$ for $D$
 2: Initialize $P$ and sort weights of each filter in $C$
 3: **for** each layer $l_i$ in $L$ from the center to the outer layers **do**
 4:     **for** $v$ in $\{5\%, 32\%\}$ **do**
 5:         $p_i = v$
 6:         $acc_p \leftarrow$ measure accuracy of $C$ with $P$ for $D$
 7:         **if** $acc - acc_p < e_t$ **then**
 8:             **break**
 9:         **else**
10:             Undo $p_i$
11:         **end if**
12:     **end for**
13: **end for**

---

When a filter applies a checkpoint, it changes the forward propagation step through the filter. Let us use an example in Fig. 1 to illustrate the MAC process of a filter which is applied a checkpoint. First, the weights and the corresponding inputs are sorted in the decreasing order of weight magnitudes. Then, the MAC operations with larger weight magnitudes are conducted first, until the checkpoint iteration is reached. If the intermediate result is less than $0$, the MAC process terminates and outputs the intermediate result directly; otherwise, it continues to complete all the MAC operations and outputs the result.

## 4.2 STEP 2: FINE-TUNE PARAMETERS

The optimization in Step 1 leads to accuracy drop due to the saved MAC operations. Thus, we further fine-tune the model to fit the applied checkpoints. The fine-tuning process is similar to the training process. The main difference is that the fine-tuning process starts with the pre-learned weights and applies the checkpoints in the forward propagation.

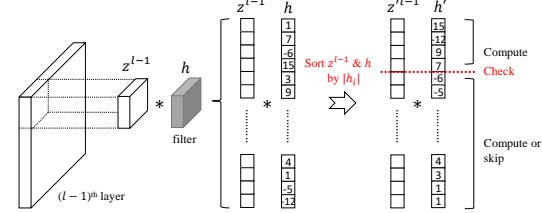

Figure 1: Example of illustrating the MAC process of a filter with a checkpoint.

The overall flow of the proposed method is as follows: Given a pre-trained CNN model $C$, the training data, and a user-defined parameter $e_t$, we first sort the weights of each filter in $C$ in the decreasing order of magnitudes. Then, we determine the checkpoint of each convolutional layer in $C$. The accuracy drop of $C$ due to the applied checkpoints cannot exceed $e_t$. Finally, we fine-tune $C$ to recover its accuracy.

### 4.3 ANALYSIS

#### 4.3.1 EARLY TERMINATING OF MAC PROCESSES

As mentioned above, the proposed method takes the advantage of the fact that many computations of the convolutional filters are unnecessary, because only the positive and large enough outputs can pass through the following activation and max pooling layers.

The functionality of a convolutional filter has been shown in EQ 1. By flattening the filter and inserting a checkpoint into the $c^{th}$ iteration, EQ 1 can be rewritten as EQ 2.

$$u = \sum_{i=0}^{c-1} z_i^{l-1} h_i + \sum_{i=c}^{H*W*K-1} z_i^{l-1} h_i + b = u_t + \sum_{i=c}^{H*W*K-1} z_i^{l-1} h_i + b \qquad (2)$$

In EQ 2, $\sum_{i=0}^{c-1} z_i^{l-1} h_i$ represents the intermediate result we use to determine whether the MAC process should terminate early or not. Let $u_t$ denotes the intermediate result.

Next, let us consider the four possibilities of $u_t$ and $u$ and the error matrix. (1) True positive: both $u_t$ and $u$ are positive. In this situation, the filter completes the MAC process and does not lead to any error. (2) True negative: both $u_t$ and $u$ are negative. In this situation, the filter terminates early and we save $(H * W * K - c)$ MAC operations without introducing any error. (3) False positive: $u_t$ is positive, but $u$ is negative. In this situation, the filter completes the MAC process without saving any computation and introducing any error, because $u_t$ is positive. (4) False negative: $u_t$ is negative, but $u$ is positive. In this situation, although the filter saves $(H * W * K - c)$ MAC operations, it generates an erroneous output. After passing through the following activation layer, the error is $\sum_{k=0}^{K-1} \sum_{q=0}^{W-1} \sum_{p=0}^{H-1} z_{p,q,k}^{l-1} h_{p,q,k} + b$, i.e., $u$.

The four situations with the corresponding MAC operation counts and errors are summarized in Table 2. Both the true negative and the false negative lead to MAC operation reduction, while the false negative introduces an error of $u$.

Table 2: Error matrix analysis of the proposed method.

|  | True Positive | True Negative | False Positive | False Negative |
|---|---|---|---|---|
| Situation | $u_t \geq 0$ and $u \geq 0$ | $u_t < 0$ and $u < 0$ | $u_t \geq 0$ and $u < 0$ | $u_t < 0$ and $u \geq 0$ |
| MAC op. | $H \times W \times K + 1$ | $c + 1$ | $H \times W \times K + 1$ | $c + 1$ |
| Error | 0 | 0 | 0 | $u$ |

#### 4.3.2 WEIGHT SORTING

In the proposed method, we determine whether a convolutional filter should terminate early according to an intermediate result. The best-case scenario is that the intermediate result and the final result have the same sign: both positive or both negative. To this end, we sort the weights in the decreasing order of magnitudes before performing the MAC process. The MAC operation that results in a larger magnitude would be conducted earlier.

Sorting weights is a straightforward idea, but it is very important in the proposed method. In the motivational example in Table 1, we have a probability of more than $60\%$ of making a correct decision at the $3^{rd}$ iteration. However, without weight sorting, the probability is approximately $50\%$.

#### 4.3.3 CHECKPOINT SETUP

As mentioned above, we layer-wisely determine the checkpoints of the convolutional layers for reducing computation efforts. Since the convolutional layers near the inputs deal with lower-level features and the convolutional layers near the outputs have larger impacts on the outputs, they usually have a smaller error tolerance. Thus, we set up checkpoints starting from the center to the outer layers.

Furthermore, for a convolutional layer, it is impractical to test all the possible checkpoints and then determine the most suitable one. Basically, a checkpoint closer to the beginning of the MAC process leads to more MAC operation reduction, but a larger error. Thus, we empirically consider only two checkpoints: the $\lfloor n * 5\% \rfloor^{th}$ iteration and the $\lfloor n * 32\% \rfloor^{th}$ iteration. Several previous works (Hinton

& Van Camp (1993); Giryes et al. (2016); Salimans & Kingma (2016)) show that the majority of weight distributions in DNNs follow the Gaussian manner. Thus we use one standard deviation region (68%) and two standard deviation region (95%) to determine the two checkpoint locations, 32% and 5%. Please note that although we consider only the two checkpoints, the proposed method can be easily extended for more checkpoints.

### 4.3.4 PARAMETER FINE-TUNING

Due to the applied checkpoints, some filters might generate erroneous outputs, causing accuracy drop. Parameter fine-tuning is a process to recover the accuracy by further training the model with the applied checkpoints. Since the difference between the model output and the expected output increases, the learnable weights will be changed in the fine-tuning process to fit the checkpoints.

## 5 EXPERIMENTAL RESULTS

We implemented the proposed method within the Caffe (Jia et al. (2014)) environment developed by the BVLC (Berkeley Vision and Learning Center). The experiments were conducted on a Linux workstation that comprises of two Intel Xeon E5-2620 2.10GHz CPUs, 64GB memory, and four NVIDIA GeForce GTX 1080 Ti GPUs.

For comparison, we applied the proposed method and two filter/channel pruning-based previous methods, CP (He et al. (2017)) and PFEC (Li et al. (2016)), to two classic CNN models, CIFAR-10 example model (C10-Net) (Jia et al. (2014)) and Network in Network (NiN) (Lin et al. (2013)), with two image classification benchmark datasets, CIFAR-10 and CIFAR-100 (Krizhevsky (2009)). Each method was conducted multiple times with different parameters to comprehend the comparison. For our method, we changed the parameter $e_t$. Ten different $e_t$, 5%, 10%, ..., 45%, and 50%, were applied. For CP and PFEC, we modified the numbers of pruned filters.

The CIFAR-10 and the CIFAR-100 datasets have 10 and 100 classes, respectively. Each dataset has 60 000 labeled images with the size of $32 \times 32$ pixels. In the experiments, we applied zero padding to each image, such that each image has a size of $40 \times 40$ pixels. Additionally, several image data augmentation techniques, including shuffling, random cropping, mirroring, and image mean, supported by Caffe were applied. When fine-tuning the weights, we conducted 20 epochs for the CIFAR-10 dataset and 40 epochs for the CIFAR-100 dataset.

### 5.1 C10-NET

#### 5.1.1 CIFAR-10 DATASET

C10-Net has three convolutional layers and two fully connected layers. The experimental results on the inference accuracy and the MAC operation count of each method under the CIFAR-10 dataset are summarized in Figure 2a. The Baseline (marked as a blue diamond) is the original model without any optimization, which conducts approximately 12.4 millions of MAC operations and achieves an accuracy of 86.53%. The red dots denote the results of our method. Basically, the accuracy decreases as $e_t$ increases. The gray squares and orange triangles denote the results of CP and FPEC, respectively.

The results show that all the three methods can trade off the accuracy against the MAC operation count. However, for several cases, our method and CP achieve similar accuracies but save more MAC operations, compared to FPEC. Furthermore, when we consider the objective of saving more MAC operations with less accuracy drop, our method with $e_t = 10\%$ obtains the best/largest trade-off ratio of the saved MAC operation count to the accuracy drop among all the methods. It achieves 50.22% MAC operation reduction with only an accuracy drop of 0.8%.

#### 5.1.2 CIFAR-100 DATASET

The experimental results on the CIFAR-100 dataset are summarized in Figure 2b. Similarly, CP is more effective than FPEC, and our method is competitive with CP. The best trade-off ratio is achieved by our method with $e_t = 5\%$, where 43.64% MAC operations are saved with an accuracy drop of 0.09%.

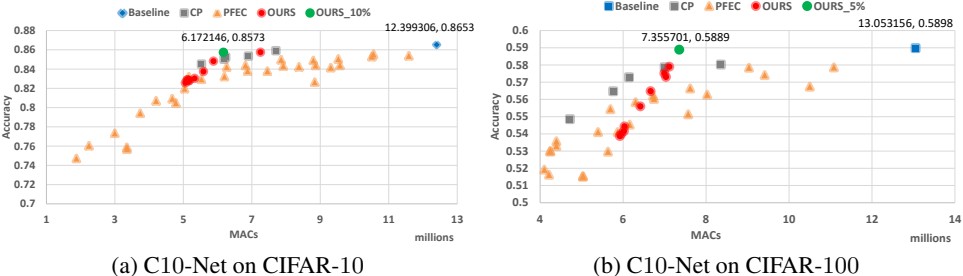

(a) C10-Net on CIFAR-10       (b) C10-Net on CIFAR-100

Figure 2: Experimental results of C10-Net on CIFAR-10 and CIFAR-100.

## 5.2 NiN

### 5.2.1 CIFAR-10 DATASET

Unlike C10-Net, NiN is composed of nine convolutional layers. The experimental results on the CIFAR-10 dataset are summarized in Figure 3a. It is observed that our method is more effective than FPEC as well as CP. The best trade-off ratio is achieved by our method with $e_t = 5\%$, where $47.43\%$ MAC operations are saved with an accuracy drop of $0.41\%$

### 5.2.2 CIFAR-100 DATASET

The experimental results on the CIFAR-100 dataset are summarized in Figure 3b. Our method is also competitive with CP and more effective than FPEC. The best trade-off ratio of our method is achieved by setting $e_t = 5\%$, where $47.73\%$ MAC operations are saved with an accuracy drop of $0.58\%$.

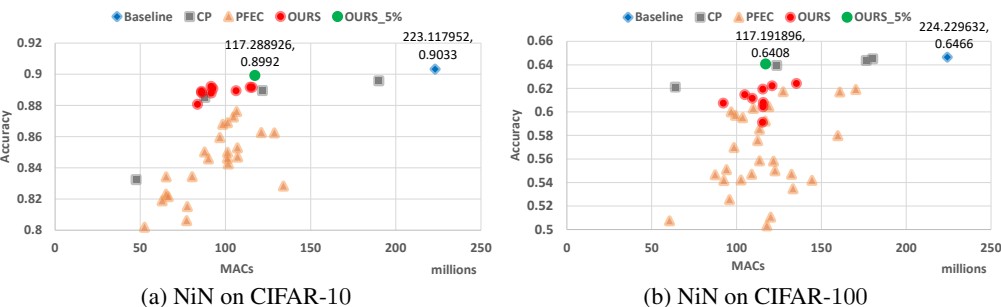

(a) NiN on CIFAR-10       (b) NiN on CIFAR-100

Figure 3: Experimental results of NiN on CIFAR-10 and CIFAR-100.

In summary, our method can save approximately $50\%$ MAC operations with an accuracy drop of less than $1\%$ for each model and dataset. Like CP and FPEC, our method can trade off the accuracy against the MAC operation count. The best trade-off ratio of our method is achieved by setting $e_t = 5\%$ or $e_t = 10\%$. Compared to CP and FPEC, our method achieves a better trade-off ratio for the CIFAR-10 dataset, and is competitive with CP for the CIFAR-100 dataset.

## 6 CONCLUSION

We present a dynamic optimization method to CNN models for reducing MAC computations in the convolutional layers. The method consists of two steps. The first step applies a checkpoint to each convolutional filter, which is used to determine whether a MAC process of the filter could terminate early according to an intermediate result during the inference phase. The second step is to fine-tune the weights of the filters to recover the accuracy drop due to the applied checkpoints. The experimental results show that the proposed method is effective for saving MAC operations with only slight accuracy drop. The proposed method could promisingly make more AI applications run on edge devices.

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
