# OpenReview forum: "Dynamic Early Terminating of Multiply Accumulate Operations for Saving Computation Cost in Convolutional Neural Networks"
_ICLR.cc/2019/Conference_

### Official Review · AnonReviewer3 · 2018-10-28
**Simple and effective method which might still have room to improve**

**Rating:** 6
**Confidence:** 3

**Review:**

In convolutional neural networks, a convolutional filter conducts a series of multiply-accumulate (MAC) operations, which is computationally heavy. To save computational cost, this manuscript proposes an algorithm to set a checkpoint in the MAC process to determine whether a filter could terminate early based on the intermediate result. The results empirically demonstrate that the proposed algorithm could save lots of MAC computations without losing too much accuracy.

Significance wise, the results look promising, though it is not always the best method to preserve accuracy while doing the pruning. There are a few experiments where CP is better at preserving accuracy. In addition, it would be great if the manuscript could also compare with other compression methods like low-rank factorization and knowledge distillation. In this manuscript, the best results are roughly 50% MAC savings (which is not equal to the saving of the whole networks) with some accuracy drop. It seems that knowledge distillation could often make the model half size without losing accuracy if properly tuned.

Quality wise, there seems to be some room to improve from a technical point of view. The proposed algorithm always starts from the center layers rather than shallow or deep layers based on the intuition that center layers have minimal influence on final output compared with other layers. This sounds about right, but lacks data support. In addition, even this intuition is correct, it might not be the optimal setting for pruning. We shall start from the layer where the ratio of accuracy drop by reducing one MAC operation is minimal. As a counterexample, if the shallow layer is the most computationally heavy, saving 10% MAC ops might save more computational cost than saving 5% in the center layer while preserving the same accuracy level.

The above sub-optimality might be further magnified by the fact that the proposed algorithm is greedy because one layer might use up all the accuracy budget before moving on to the next candidate layer as shown in line 7 of Algorithm 1.

The method is quite original, and the manuscript is very well written and easy to follow.

---

> ### Author Response · Authors · 2018-11-19
> **Author’s response to AnonReviewer3’s comments**
>
> We appreciate the valuable comments and the efforts of the reviewer on this manuscript. The detailed remarks are very helpful for improving the manuscript. We have made necessary changes to respond to all the comments.
>
> Q1: In convolutional neural networks, a convolutional filter conducts a series of multiply-accumulate (MAC) operations, which is computationally heavy. To save computational cost, this manuscript proposes an algorithm to set a checkpoint in the MAC process to determine whether a filter could terminate early based on the intermediate result. The results empirically demonstrate that the proposed algorithm could save lots of MAC computations without losing too much accuracy.
>
> A1: We appreciate this comment.
>
> Q2: Significance wise, the results look promising, though it is not always the best method to preserve accuracy while doing the pruning. There are a few experiments where CP is better at preserving accuracy. In addition, it would be great if the manuscript could also compare with other compression methods like low-rank factorization and knowledge distillation. In this manuscript, the best results are roughly 50% MAC savings (which is not equal to the saving of the whole networks) with some accuracy drop. It seems that knowledge distillation could often make the model half size without losing accuracy if properly tuned.
>
> A2: We appreciate this comment. In fact, CP uses low-rank factorization as one of its key techniques. Although we do not compare to a low-rank factorization-based method, our approach should be competitive as well. Furthermore, we do not compare to knowledge distillation, because our approach is compatible with it. They could be combined to achieve better effectiveness.  For example, one can apply our approach to the student network obtained from the knowledge distillation method. We will try to conduct some experiments to see if the combination is promising.
>
> Q3: Quality wise, there seems to be some room to improve from a technical point of view. The proposed algorithm always starts from the center layers rather than shallow or deep layers based on the intuition that center layers have minimal influence on final output compared with other layers. This sounds about right, but lacks data support. In addition, even this intuition is correct, it might not be the optimal setting for pruning. We shall start from the layer where the ratio of accuracy drop by reducing one MAC operation is minimal. As a counterexample, if the shallow layer is the most computationally heavy, saving 10% MAC ops might save more computational cost than saving 5% in the center layer while preserving the same accuracy level.
>
> A3: We appreciate this comment. We are trying to develop a new method for setting checkpoints based on the suggestion. The new method has a pre-process that analyzes the error tolerance of each layer, and then it sets checkpoints starting from the layer with the highest error tolerance. Currently, we are looking for a good method to estimate the error tolerance of a layer and conducting some experiments to show its effectiveness. When the experimental results are ready, we will post the results onto the website.
>
> Q4: The above sub-optimality might be further magnified by the fact that the proposed algorithm is greedy because one layer might use up all the accuracy budget before moving on to the next candidate layer as shown in line 7 of Algorithm 1.
>
> A4: We appreciate this comment. We agree with the reviewer that the proposed method for setting checkpoints is greedy and it may suffer from the mentioned issue, although the issue did not happen in the considered cases. We have adopted the reviewer’s suggestion to present a new method. It should be able to mitigate or solve the issue.
>
> Q5: The method is quite original, and the manuscript is very well written and easy to follow.
>
> A5: We appreciate this comment.

---

### Official Review · AnonReviewer1 · 2018-11-02
**Unclear if it will save wall clock time.**

**Rating:** 6
**Confidence:** 5

**Review:**

This paper motivates itself by observing that not all convolutional weights are required to make an accurate prediction. In the introduction the authors envision a system similar to a cascaded classifier [Viola and Jones 2001] (I draw this conclusion not the paper). However the wording of the introduction is not clear or it does not align with what is presented in the paper.

The approach in the paper does not perform early stopping dynamically during the feedforward phase. The approach removes weights which do not impact the accuracy after training has completed and the fine tunes the resulting network.

The clarity of the introduction must be addressed however the work is still interesting. I recommend the authors try to make the introduction as accessible as possible.


Also there are very general statements like "The activation layer introduces non-linearity into the system for obtaining better accuracy." which do not contribute to the message of the paper. The paper will be fine without these statements. Shorter is better.

Section 3 is good as a motivating example. However the conclusion “Thus, our focus is to develop an effective method of choosing a good intermediate result for saving more MAC operations with less accuracy drop” is not very clear. More insights written hear would be better.

One major flaw is that no analysis with respect to time of computation was performed. GPUs offer the advantage of being optimized for convolutions so it is possible that there is no speedup. Because of this it is unclear if the method would save time. The results clearly show individual computations (MACs) are reduced but it is not clear how this correlates with wall clock time.

Why do you start with the centre layers? I understand the heuristic you’re using, that the middle layers won’t have high or low-level features, and that they won’t break the other layers as badly if you modify them, but I feel like this is core to your method and it’s not adequately justified. I’d like to see some experiments on that and whether it actually matters to the outcome. Also, you don’t say if you start at the middle then go up a layer, or down a layer. I think this matters since your proposed C10-Net has only 3 convolutional layers.

All the filters in the same layer share a common checkpoint. Is that a good idea? What is the cost of doing this on a per-filter level? What is the cost on a per-layer level? Discussing runtime estimates for the added computation at training would make it more clear what the cost of the method is.
In section 4.4.3. you mention that the majority of weight distributions in CNNs follow the Gaussian manner. Could you cite something of this? You might also want to move that in Step 1 (section 4.1.), since it seems to be your motivation for selection of checkpoint locations (5% and 32%) and I had no idea why you selected those values at that point.

Typos:
Typo on page 3: “exploits redundancies inter feature maps to prune filters and feature maps”

Structural:
Maybe section 4.3 should be part of the description of Section 4. Proposed Method, not its own subsection.

Maybe table 2 should go at the end of section 4.4.1, because you describe it the error and columns in section 4.4.1.

---

> ### Author Response · Authors · 2018-11-20
> **Author’s response to AnonReviewer1’s comments - 1**
>
> We appreciate the valuable comments and the efforts of the reviewer on this manuscript. The detailed remarks are very helpful for improving the manuscript. We have made necessary changes to respond to all the comments.
>
> Q1: This paper motivates itself by observing that not all convolutional weights are required to make an accurate prediction. ...
>
> A1: We appreciate this comment. In fact, the technique “early terminating” proposed in this paper terminates a MAC process before it completes for saving computation cost. The technique is different from that used in [1][2] which early terminates the whole network and makes a prediction directly at an early layer. We have revised the introduction section on Page 2 to clarify the differences as suggested.
>
> [1] Viola, Paul, and Michael Jones. "Rapid object detection using a boosted cascade of simple features." Computer Vision and Pattern Recognition, 2001. CVPR 2001. Proceedings of the 2001 IEEE Computer Society Conference on. Vol. 1. IEEE, 2001.
> [2] Teerapittayanon, Surat, Bradley McDanel, and H. T. Kung. "Branchynet: Fast inference via early exiting from deep neural networks." Pattern Recognition (ICPR), 2016 23rd International Conference on. IEEE, 2016.
>
> Q2: The approach in the paper does not perform early stopping dynamically during the feedforward phase. ...
>
> A2: We appreciate this comment. We say that the proposed technique is dynamic because whether a MAC process early stops or not is related to the input data. That is, a MAC process may early stop for one input image, but does not for another input image.  In addition, we do not remove weights, because the criticality of a weight differs for different input images. As suggested, we have revised the introduction section on Page 2 to clarify the proposed method.
>
> Q3: The clarity of the introduction must be addressed however the work is still interesting. ...
>
> A3: We appreciate this comment. As suggested, we have revised the introduction section on Page 2 to clarify the proposed method and clearly stated the differences of the proposed method compared with previous methods.
>
> Q4: Also there are very general statements like "The activation layer introduces non-linearity into the system for obtaining better accuracy." ...
>
> A4: We appreciate this comment. We have revised the paper as suggested.
>
> Q5: Section 3 is good as a motivating example. ...
>
> A5: We appreciate this comment. We originally would like to emphasize that the proposed idea is promising, and our next step is to address the issue of how to set good checkpoints which lead to more MAC operation saving with less accuracy drop. We have revised the statements to clarify our intention.
>
> Q6: One major flaw is that no analysis with respect to time of computation was performed. ...
>
> A6: We appreciate this comment. We agree that wall-clock-time is an important performance criteria. However, as the reviewer mentioned, current GPUs have been well optimized for convolution operations. Thus, the proposed method actually does not save execution time, but instead spends more time. In our implementation, we design a new function to replace the Cuda function CuBLAS SGEMM, use a counter for checking whether the checkpoint is reached, and use a look-up table to record the indexes of the sorted weights. These modification and extra effort cause the main time overhead.
> We think that the main benefit of the proposed method is that we can turn off the unused GPU nodes or threads to save power/energy consumption. Since the proposed method is currently executed on a workstation, it is difficult to measure the saved power/energy consumption. Thus, we use MAC operations as the measuring criteria in the experiments. One possible way to address the issue is to port the programs onto a GPU development board, such as NVIDIA Jetson TX2, so that we are able to measure the consumed power/energy. This will be our future work.
> We have conducted a simple experiment to demonstrate the time overhead of the proposed method. We used the optimized (ours) and the non-optimized (baseline) C10-Net and NiN to test the CIFAR-10 testing data and recorded the average execution time of 200 input images. The following table summarizes the results comparing the baseline and ours.
>
> ----------------------------------------------------------------
>                |C10-Net, et = 10%|     NiN, et = 5%     |
> ----------------------------------------------------------------
> Baseline|      116.554 ms     |       879.404 ms     |
> ----------------------------------------------------------------
> Ours      |       585.659 ms     |       6141.77 ms     |
> ----------------------------------------------------------------

---

> ### Author Response · Authors · 2018-11-20
> **Author’s response to AnonReviewer1’s comments - 2**
>
> Q7: Why do you start with the centre layers? ...
>
> A7: We appreciate this comment. The original heuristic is based on our inference that the convolutional layers near the inputs deal with lower-level features and the convolutional layers near the outputs have larger impacts on the outputs, and thus they usually have a smaller error tolerance. Therefore, we start from the center layer, and then go to the layer near the input and then the layer the output. We have clarified the heuristic on Page 5 in the revised paper.
> To show the effectiveness of the proposed heuristic, we further applied two additional heuristics: one starts from the first layer to the last layer and the other one starts from the last layer to the first layer. The experimental results on the set checkpoint location for each convolutional layer, the network accuracy and the conducted MAC count are shown in the following tables. In the experiments, the parameter et is set to 10% for C10-Net and 5% for NiN, and the benchmark dataset is the CIFAR-10 dataset.
> For C10-Net, the experimental results show that the all the three heuristics lead to the same results. The reason could be that C10-Net is too small, so that the optimization order does not affect the results.
>
> C10-Net, et = 10%
> ---------------------------------------------------------------------------------
>                                 |Conv1|Conv2|Conv3|Accuracy| MACs |
> ---------------------------------------------------------------------------------
> Baseline	                |     -     |      -    |     -     |  86.53% |12.40M|
> ---------------------------------------------------------------------------------
> Ours (original)      |  32% |  32%  |   5%   |  85.73%  | 6.17M |
> ---------------------------------------------------------------------------------
> Ours (first → last) |  32% |  32%  |   5%   |  85.73%  | 6.17M |
> ---------------------------------------------------------------------------------
> Ours (last → first) |  32% |  32%  |   5%   |  85.73%  | 6.17M |
> ---------------------------------------------------------------------------------
>
> Furthermore, for NiN, the experimental results show that the optimization order affects the final results. The original method and the heuristic starting from the last layer to the first layer lead to the same results. They also achieve a better trade-off ratio, compared with the heuristic starting from the first layer to the last layer.
>
> NiN, et = 5%
> --------------------------------------------------------------------------------------------------------------------------------------------
>                              |Conv1|Cccp1|Cccp2|Conv2|Cccp3|Cccp4|Conv3|Cccp5|Cccp6|Accuracy |   MACs  |
> --------------------------------------------------------------------------------------------------------------------------------------------
> Baseline               |    -     |     -    |     -     |     -     |     -    |     -    |     -     |    -     |    -     |   90.33%  |223.12M|
> --------------------------------------------------------------------------------------------------------------------------------------------
> Ours (original)    |  32% |  32% |   5%   |  32%  |   5%  |   5%  |   5%   |   5%  |   5%  |   89.92%  |117.29M|
> --------------------------------------------------------------------------------------------------------------------------------------------
> Ours (first→last) |   5%  |  32% |   5%   |  32%  |  32% |  32% |   5%   |   5%  |   5%  |  89.27%  |113.75M|
> --------------------------------------------------------------------------------------------------------------------------------------------
> Ours (last→first) |  32% |  32% |   5%   |  32%  |  5%   |   5%  |   5%   |   5%  |   5%  |  89.92%  |117.29M|
> --------------------------------------------------------------------------------------------------------------------------------------------
>
> We believe that the optimization order is an important factor determining the effectiveness of the proposed method. The current method starting from the center to the outer layers should have some room to improve. Thus, we will keep developing a more  effective method.
> On the other hand, we would like to appreciate AnonReviewer3 who suggests a new heuristic, which starts from the layer that has a highest error tolerance to the layer with the lowest error tolerance. The new method has a pre-process that analyzes the error tolerance of each layer, and then it sets checkpoints starting from the layer with the highest error tolerance. Currently, we are looking for a good method to estimate the error tolerance of a layer and conducting some experiments to show its effectiveness. When the experimental results are ready, we will post the results onto the website.

---

> ### Author Response · Authors · 2018-11-20
> **Author’s response to AnonReviewer1’s comments - 3**
>
> Q8: All the filters in the same layer share a common checkpoint. ...
>
> A8: We appreciate this comment. To set a checkpoint, we currently apply all the images in the training data set to check if it is applicable. If there are n layers, in the worst case, we need n*2 (2 possible checkpoint locations) rounds of testing all the training images. However, if we set checkpoints on the per-filter level, we need m*2 rounds, where m is the filter count which is usually far larger than n. For example, the C10-Net has three convolutional layers (n=3) and 128 filters (m=128). We believe that setting checkpoints on the per-filter level should be more effective, because the error tolerance of each filter should be different. However, it is very time-expensive and could be unaffordable for large networks. One possible solution to address the issue is to reduce the applied images in each round, while it may affect the measured accuracy drop. We have clearly stated why determining checkpoints on a per-filter level is very time-expensive on Pages 4 and 5 in the revised paper.
> As for the concern on weight distributions, some works [1-3] claim that the weight distributions in DNNs can be formulated as a Normal/Gaussian distribution. Additionally, the Batch Normalization method [4] uses Gaussian-initialized weights to reinforce DNN training. We also conducted normal test with p-value 10e-3 and found that most weight distributions pass through the hypothesis region. Thus, we believe that the statement that the majority of weight distributions in CNNs follow the Gaussian manner is reasonable.
> Based on the Gaussian distribution of weights, we then use two standard deviation points to determine the checkpoints. Because our method deals with larger-magnitude weights first, we use the second standard deviation point as the first possible checkpoint location (100%-95%=5%). Furthermore, we use the first standard deviation point as the second possible checkpoint location (100%-68% = 32%).
> Basically, the proposed method is a heuristic and we believe that there must exist other methods (e.g., using other checkpoint locations or more checkpoints) which could be more effective. However, the effectiveness of the heuristic have been demonstrated by the experiments. In the future, we will try to develop a smarter method (maybe a machine learning-based method) to determine the checkpoint locations.
>
> [1] Hinton, Geoffrey E., and Drew Van Camp. "Keeping the neural networks simple by minimizing the description length of the weights." Proceedings of the sixth annual conference on Computational learning theory. ACM, 1993.
> [2] Salimans, Tim, and Diederik P. Kingma. "Weight normalization: A simple reparameterization to accelerate training of deep neural networks." Advances in Neural Information Processing Systems. 2016.
> [3] Giryes, Raja, Guillermo Sapiro, and Alexander M. Bronstein. "Deep neural networks with random gaussian weights: A universal classification strategy?." IEEE Trans. Signal Processing 64.13 (2016): 3444-3457.
> [4] Ioffe, Sergey, and Christian Szegedy. "Batch normalization: Accelerating deep network training by reducing internal covariate shift." arXiv preprint arXiv:1502.03167 (2015).
>
> Q9: Typo on page 3: “exploits redundancies inter feature maps to prune filters and feature maps”
>
> A9: We appreciate this comment. We have corrected the typo in the revised version.
>
> Q10: Maybe section 4.3 should be part of the description of Section 4. Proposed Method, not its own subsection.
>
> A10: We appreciate this comment. We will adopt the suggestion to merge Section 4.3 into Section 4.
>
> Q11: Maybe table 2 should go at the end of section 4.4.1, because you describe it the error and columns in section 4.4.1.
>
> A11: We appreciate this comment. We will move Table 2 to the suggested location.

---

### Official Review · AnonReviewer4 · 2018-11-13
**Interesting work but some points need clarification.**

**Rating:** 5
**Confidence:** 3

**Review:**

This paper proposes a new method for speeding up convolutional neural networks. Different from previous work, it uses the idea of early terminating the computation of convolutional layers. The method itself is intuitive and easy to understand. By sorting the parameters in a descending order and early stopping the computation in a filter, it can reduce the computation cost (MAC) while preserving accuracy.

1. The networks used in the experiments are very simple. I understand that in the formulation part the assumption is that ReLU layer is put directly after convolutional layer. However, in state-of-the-art network, batch normalization layer is put between convolutional layer and ReLU non-linearity. It would add much value if the authors could justify the use cases of the proposed method on the widely adopted networks such as ResNet.

2. I notice that there is a process that sort the parameters in the convolutional layers. However, the authors do not give any time complexity analysis about this process. I would like see how weight sorting influences the inference time.

3. The title contains the word “dynamic”. However, I notice that the parameter e used in the paper is predefined (or chosen from a set predefined of values). So i am not sure it is appropriate to use the word “dynamic” here. Correct me if i am wrong here.

4. In the experiment part, the authors choose two baselines: FPEC [1]. However, to my knowledge, their methods are performed on different networks. Also, the pruned models from their methods are carefully designed using sensitivity analysis. So I am curious how are the baselines designed in your experiments.

Overall this paper is well-written and points a new direction to speeding up neural networks. I like the analysis in section 3.

I will consider revising the score if the authors can address my concerns.


[1] Pruning Filters for Efficient ConvNets. Li et al., ICLR 2017.

---

> ### Author Response · Authors · 2018-11-20
> **Author’s response to AnonReviewer4’s comments**
>
> We appreciate the valuable comments and the efforts of the reviewer on this manuscript. The detailed remarks are very helpful for improving the manuscript. We have made necessary changes to respond to all the comments.
>
> Q1: This paper proposes a new method for speeding up convolutional neural networks. ...
>
> A1: We appreciate this comment.
>
> Q2: The networks used in the experiments are very simple. ...
>
> A2: We appreciate this comment. We think that our method could be extended to deal with the mentioned network. Our current method compare the intermediate result at the checkpoint to 0 to determine whether a MAC process should stop or not. If the result is less than 0, we terminate the MAC process. To our understanding, the batch normalization method normalizes the outputs of the convolutional layers according to four learned parameters, mu, sigma, gamma, and beta, and then passes the results to the following activation layers. One possible solution to address the issue is to apply batch normalization to the intermediate result before it is compared to 0. Another possible solution is to compare the intermediate result to a non-zero value. The value is determined according to the four learned parameters, so that when the intermediate result is less than the value, it implies that the normalized intermediate result will be less than 0. Additionally, since the final fine-tuning process updates the four learned parameters, the value to be compared would be updated as well.
>
> Q3: I notice that there is a process that sort the parameters in the convolutional layers. ...
>
> A3: We appreciate this comment. As AnonReviewer1 mentioned, current GPUs have been well optimized for convolution operations. However, to implement the proposed technique, we need to design a new function to replace the Cuda function CuBLAS SGEMM for MACs. Thus, our method actually does not speed up the inference time. In our implementation, we use a look-up table to record the indexes of the sorted weights, so that we only need to sort the weights of a filter once. In addition, we use a counter for checking whether the checkpoint is reached. These extra efforts all cause time overhead. In fact, we think that the main benefit of the proposed method should be that we can turn off the unused GPU nodes or threads to save power/energy consumption. However, since the proposed method is currently executed on a workstation, it is difficult to measure the saved power/energy consumption. Thus, we use MAC operations as the measuring criteria in the experiments.
> We have conducted a simple experiment to demonstrate the time overhead of the proposed method. We used the optimized (ours) and the non-optimized (baseline) C10-Net and NiN to test the CIFAR-10 testing data and recorded the average execution time for a batch of 200 input images. The following table summarizes the results comparing the baseline and ours.
>
> ---------------------------------------------
>                |   C10-Net,  |       NiN,     |
>                |   et = 10%  |     et = 5%  |
> ---------------------------------------------
> Baseline|116.554 ms|879.404 ms|
> ---------------------------------------------
> Ours      |585.659 ms|6141.77 ms |
> ---------------------------------------------
>
> Q4: The title contains the word “dynamic”. ...
>
> A4: We appreciate this comment. The parameter et is predefined by the user for trading off inference accuracy and MAC operations. In this paper, we use the word “dynamic” because whether a MAC process early stops or not is conditional and related to the input images. That is, a MAC process may early stop for one input image, but does not for another input image. We have revised the introduction section on Page 2 to clarify the word “dynamic”.
>
> Q5: In the experiment part, the authors choose two baselines: CP [1] and FPEC [2]. ...
>
> A5: We appreciate this comment. The programs of CP [1] and PFEC [2] we used in the experiments were downloaded from the github website [3][4]. They were released by the authors or had been confirmed that they can reproduce the experiments in the original papers [1][2]. Both the CP and the PFEC programs allow a user to determine the ratio of filters to be dropped. Thus, in our experiments, the results of CP and PFEC were obtained by applying several different ratios.
> Furthermore, in the future, we will try to apply our method to more networks and datasets, such as ResNet, to show that our extended method works well for the batch normalization operation and to have a more comprehensive comparison with CP and PFEC.
>
> [1] He, Yihui, et al. "Channel pruning for accelerating very deep neural networks." ICCV. Vol. 2. No. 6. 2017.
> [2] Li, Hao, et al. "Pruning filters for efficient convnets." arXiv:1608.08710. 2016.
> [3] CP: https://github.com/yihui-he/channel-pruning
> [4] PFEC: https://github.com/slothkong/DNN-Pruning
>
> Q6: Overall this paper is well-written and points a new direction to speeding up neural networks. ...
>
> A6: We appreciate this comment.

---

> > ### Comment · AnonReviewer4 · 2018-12-05
> > **Response**
> >
> > Thanks for the explanation and extra experiments.
> >
> > From your time measurement, it seems that the time overhead is about 4 or 5 times of the original model. I am not sure if I understand the results correctly. But from my understanding, it seems that there is not any reduced execution time but much longer. So it would be helpful if the authors can further justify why this happens or the ways to improve the execution time?
> >
> > About FPEC, the original papers use a uniform pruning ratio for most convolutional layers. So does the ratio in your experiments refer to this uniform pruning ratio?

---

> > > ### Author Response · Authors · 2018-12-13
> > > **Author’s response to AnonReviewer4’s comments**
> > >
> > > We appreciate AnonReviewer4's comments.
> > >
> > > So far, the overhead on the inference time is certainly an issue of our method. We summarize the extra efforts that our method could lead to as follows:
> > >
> > > 1. Checkpoint polling time. Our method needs to check if the checkpoint is reached at each iteration. As the result, GPUs have to spend extra efforts for the checking.
> > >
> > > 2. Weight sorting time and look-up table checking time. In our method, the MAC operation with a larger-magnitude weight is conducted first. To achieve this, we need to sort the weights before the MAC operations. Furthermore, to prevent from repeating the same weights in a filter, we use a look-up table to store the indexes of the sorted weights. As a result, our method needs to query the look-up table, which costs GPUs extra efforts.
> > >
> > > As a result, our method currently implemented on GPUs does not lead to benefit on the inference time. To mitigate the overhead, we would like to develop a dedicated hardware accelerator, such as EIE [1], to support our method in the future.
> > >
> > > As for CP and PFEC, to our understanding, they allow users to use different pruning ratios for different layers. The reported parameters on Github also show that not all the pruning ratios in the network are identical. In our experiments, since the network models under consideration are different from that used by the CP and PFEC works, we do not adopt the reported parameters. We try different combinations of pruning ratios for both CP and PFEC in the experiments. Thus, the used CP and PFEC do not use a uniform pruning ratio for all layers in our experiments. In the future, we will apply our method to the VGGNet and ResNet to show that our method could work well for other CNN models as well.
> > >
> > > [1] Han, Song, et al. "EIE: efficient inference engine on compressed deep neural network." Computer Architecture (ISCA), 2016 ACM/IEEE 43rd Annual International Symposium on. IEEE, 2016.

---

### Author Response · Authors · 2018-11-26
**A new heuristic of setting checkpoints starting from the layer that leads to the best trade-off ratio of  saved MAC operation count to accuracy drop.**

We present a new heuristic inspired by AnonReviewer3 to determine the layer order of setting checkpoints. Since our objective is to save more MAC operations with less accuracy drop. The new heuristic starts from the layer that achieves the best trade-off ratio of the saved MAC operation count to the accuracy drop when a checkpoint is applied to it. Thus, in the new heuristic, we conduct a process to compute the trade-off ratio that each layer leads to, before setting checkpoints.
The following table shows the experimental results of the new heuristic for NiN with et=5% under the CIFAR-10 dataset. “Ours (5%)” denotes the new heuristic, in which the trade-off ratio of each layer is computed based on the checkpoint of 5%. “Ours (32%)” denotes the new heuristic, in which the trade-off ratio of each layer is computed based on the checkpoint of 32%. The results show that the three different methods/orders result in the same optimization result, although the orders of setting checkpoints are different. The reason that the three methods have the same optimization result could be that the considered et is too small. We are conducting more experiments with larger et to have more comprehensive comparison.

NiN, et = 5%
--------------------------------------------------------------------------------------------------------------------------------------
                       |Conv1|Cccp1|Cccp2|Conv2|Cccp3|Cccp4|Conv3|Cccp5|Cccp6|Accuracy|  MACs   |
--------------------------------------------------------------------------------------------------------------------------------------
Baseline        |     -     |     -    |     -    |     -     |     -     |     -    |     -     |     -    |     -    |  90.33%  |223.12M|
--------------------------------------------------------------------------------------------------------------------------------------
Ours (ori.)    |  32%   |  32% |  5%  |   32% |   5%    |   5%  |   5%   |   5%  |   5%  | 89.92%  |117.29M|
Order	       |     8     |    6    |    4   |     2     |     1     |     3   |     5    |    7    |     9   |
 -------------------------------------------------------------------------------------------------------------------------------------
Ours (5%)     |  32%   |  32% |  5% |   32%  |    5%   |  5%   |  5%    |   5%  |   5%  | 89.92%  |117.29M|
Order	       |     9     |    8     |  4    |    7      |    5      |    6    |     3    |     1    |    2    |
--------------------------------------------------------------------------------------------------------------------------------------
Ours (32%)   |  32%   |  32% |  5% |   32%  |    5%   |  5%   |  5%    |   5%  |   5%  | 89.92%  |117.29M|
Order	       |     9     |    8     |  2    |    7      |    3      |    4    |     1    |     5    |    6    |
--------------------------------------------------------------------------------------------------------------------------------------

Furthermore, we also tried two heuristics that start from the layer with minimal accuracy drop and start from the layer with maximal accuracy drop. The accuracy drop is estimated based on the checkpoint of 5%. However, the results show that the achieved trade-off ratio is not as good as the new heuristic. Thus, we think that the new heuristic should be more promising.

NiN, et = 5%
-----------------------------------------------------------------------------------------------------------------------------------------------
                                   |Conv1|Cccp1|Cccp2|Conv2|Cccp3|Cccp4|Conv3|Cccp5|Cccp6|Accuracy|  MACs   |
-----------------------------------------------------------------------------------------------------------------------------------------------
Baseline                    |      -    |     -    |     -    |     -     |     -     |     -    |     -     |     -    |     -    |  90.33%  |223.12M|
-----------------------------------------------------------------------------------------------------------------------------------------------
Ours(min → max)   |    5%  |  32% |   5%  |  32%  |   32% |  32% |   5%   |   5%  |   5%  |  89.44%  |113.73M|
Order	                   |    7     |    8     |    4    |    9     |      5   |     6    |     1    |    2    |    3    |
-----------------------------------------------------------------------------------------------------------------------------------------------
Ours (max → min)  |  32%  |  5%   |   5%  |  32%  |   32% | 32%  |  32%  |  32% |   5%  |  89.43%  |113.33M|
Order	                   |     3    |    2     |    6    |    1     |     4    |    5     |    7     |    8    |    9   |
-----------------------------------------------------------------------------------------------------------------------------------------------

---

### Meta-Review · Area_Chair1 · 2018-12-17
**no practical speedup**

**Confidence:** 4
**Recommendation:** Reject

**Metareview:**

This paper proposes a new method for speeding up convolutional neural networks. It uses the idea of early terminating the computation of convolutional layers. It saves FLOPs, but the reviewers raised a critical concern that it doesn't save wall-clock time. The time overhead is about 4 or 5 times of the original model. There is not any reduced execution time but much longer. The authors agreed that "the overhead on the inference time is certainly an issue of our method". The work is not mature and practical. recommend for rejection.